# Canrenone Restores Vasorelaxation Impaired by Marinobufagenin in Human Preeclampsia

**DOI:** 10.3390/ijms23063336

**Published:** 2022-03-19

**Authors:** Natalia I. Agalakova, Yulia N. Grigorova, Ivan A. Ershov, Vitaly A. Reznik, Elena V. Mikhailova, Olga V. Nadei, Leticia Samuilovskaya, Larisa A. Romanova, C. David Adair, Irina V. Romanova, Alexei Y. Bagrov

**Affiliations:** 1Sechenov Institute of Evolutionary Physiology and Biochemistry, 194223 St. Petersburg, Russia; nagalak@mail.ru (N.I.A.); yulia.grigorova2@nih.gov (Y.N.G.); elenamikhailova87@gmail.com (E.V.M.); olganadej@gmail.com (O.V.N.); letitcia0311@gmail.com (L.S.); romanova.irina@gmail.com (I.V.R.); 2Department of Obstetrics and Gynecology, St. Petersburg State Pediatric Medical University, 194100 St. Petersburg, Russia; ershov_ia@inbox.ru (I.A.E.); vitaliy-reznik@mail.ru (V.A.R.); l_romanova2011@mail.ru (L.A.R.); 3Department of Obstetrics and Gynecology, University of Tennessee, Chattanooga, TN 37341, USA; david@solasbio.com; 4Padakonn Pharma, 20309 Narva, Estonia

**Keywords:** preeclampsia, Na/K-ATPase, cardiotonic steroids, marinobufagenin, Fli1, sodium chloride, TGFβ, vascular stiffness

## Abstract

Previous studies implicated cardiotonic steroids, including Na/K-ATPase inhibitor marinobufagenin (MBG), in the pathogenesis of preeclampsia (PE). Recently, we demonstrated that (i) MBG induces fibrosis in rat tissues via a mechanism involving Fli1, a negative regulator of collagen-1 synthesis, and (ii) MBG sensitive Na/K-ATPase inhibition is reversed by mineralocorticoid antagonists. We hypothesized that in human PE elevated MBG level is associated with the development of fibrosis of the umbilical arteries and that this fibrosis can be attenuated by canrenone. Fifteen patients with PE (mean BP = 118 ± 4 mmHg; 34 ± 2 years; 38 ± 0.3 weeks gest. age) and twelve gestational age-matched normal pregnant subjects (mean BP = 92 ± 2 mmHg; 34 ± 1 years; 39 ± 0.2 weeks gest. age) were enrolled in the study. PE was associated with a higher plasma MBG level, with a four-fold decrease in Fli1 level and a three-fold increase in collagen-1 level in the PE umbilical arteries vs. those from the normal subjects (*p* < 0.01). Isolated rings of umbilical arteries from the subjects with PE exhibited impaired responses to the relaxant effect of sodium nitroprusside vs. control vessels (EC_50_ = 141 nmol/L vs. EC_50_ = 0.9 nmol/L; *p* < 0.001). The effects of PE on Fli1 and collagen-1 were blocked by the in vitro treatment of umbilical arteries by 10 μmol/L canrenone. Similar results were obtained for umbilical arteries pretreated with MBG. These data demonstrate that elevated MBG level is implicated in the development of the fibrosis of umbilical arteries in PE, and that this could be blocked by mineralocorticoid antagonists.

## 1. Introduction

Marinobufagenin (MBG) is a bufadienolide inhibitor of the alpha-1 subunit of the Na/K-ATPase, the main isoform in the vascular wall and in the kidney [1,2]. In mammals MBG is a natriuretic [3], it regulates cardiac contractility [4] and vascular tone [5]. Recent evidence suggests that MBG suppresses Fli1, a nuclear transcription factor, and a negative regulator of the COL-1 gene, and/or causes activation of the TGFβ-dependent signaling pathway, which leads to the synthesis of collagen [5,6]. Accordingly, in hypertensive humans, MBG correlates with vascular stiffness [7,8]. Another example of the harmful effects of MBG is preeclampsia (PE), which complicates up to 5% of pregnancies [5,9]. In PE elevated MBG levels are associated with increased blood pressure [10,11] and with the development of fibrosis and stiffness of the umbilical arteries [5,12,13]. As a result, in PE, due to elevated MBG, umbilical arteries lose their capacity to relax although their contractile response remains unchanged [5]. Therefore, in PE patients, heightened MBG plasma levels represent a potential target for therapy and for the immunoneutralization of MBG [14]. We developed an anti-MBG monoclonal antibody that possesses a remarkable capacity to reduce blood pressure and cardiac fibrosis in rats with renal failure [11,15]. In rats with experimental PE, this antibody in vivo reduced blood pressure by 40 mmHg and restored Na/K-ATP activity in the aorta [15].

Recently, we demonstrated that canrenone is able to antagonize Na/K-ATPase inhibitory and pro-fibrotic effects of MBG in rat vascular tissue [16]. Because previously spironolactone and canrenone, its active metabolite, were shown to reduce arterial stiffness in patients with chronic renal disease [17,18] and act as an Na/K-ATPase antagonist [16] we undertook the present study. The goals of the present experiments were to assess an ability of mineralocorticoid antagonist canrenone to (i) counterbalance the fibrotic effect of MBG and (ii) to define which of the mechanisms, Fli1-dependent or TGF-β-dependent, is responsible for the stiffening and fibrosis of umbilical arteries in PE. 

## 2. Results

Fifteen patients with PE (age, 34 ± 2 years; gestational age, 37.9 ± 0.2 weeks) and twelve normotensive pregnant subjects [19] (age, 34 ± 1 years; gestational age, 39 ± 1 weeks) were enrolled in the study (Table 1). Patients with PE exhibited an increase in systolic and diastolic blood pressures, and increased protein excretion (Table 1).

An analysis of placenta sections with hematoxylin and eosin staining of the chorionic villi of healthy patients demonstrated rounded arteries with well-defined muscular shell, while in patients with PE, the chorionic villi have elongated, flattened arteries (Figure 1A,B). 

An evaluation of the MBG immunopositive material in the placenta of the control group demonstrates the presence of structures both in the cytotrophoblast cells and in the connective tissue cells of the chorionic (Figure 1C,D). In patients with PE, the amount of MBG-immunopositive material was increased (Figure 1D), the optical density of MBG in cytotrophoblast was 0.25 ± 0.02 a.u. in PE vs. 0.14 ± 0.01 a.u. in control samples (*p* < 0.05). Plasma and placental levels of MBG immunoreactivity in patients with PE were significantly higher than the control group, while in placenta the levels of mRNA Fli1 were decreased four-fold in PE versus control samples (Figure 2).

As presented in Figure 3, development of PE was associated with a four-fold reduction in the expression of umbilical arteries Fli1, and, conversely, level of collagen-1 was significantly elevated vs. tissue from normal pregnancy. Figure 3B–D demonstrates absence of the changes in TGF-β and collagen-4 in umbilical arteries from patients with PE. As shown in Figure 4, rings from umbilical arteries from normal pregnant subjects pretreated with 10 nmol/L sodium nitroprusside exhibited 50% relaxation following submaximal (100 nmol/L) concentration of endothelin-1 (EC_50_ = 58 ± 14 nmol/L). The responsiveness of the vascular rings obtained from patients with PE was markedly reduced (EC_50_ = 3 ± 5 μmol/L) as compared to the control rings. In the presence of canrenone umbilical arteries restored their ability to relax following SNP addition (EC_50_ = 40 ± 7 nmol/L). 

Next, we studied whether ex vivo incubation of the explants of umbilical arteries from subjects with non-complicated pregnancies in the presence of nanomolar concentrations of MBG would mimic the effects of preeclampsia. As presented in Figure 5, 24 h incubation of umbilical artery rings in the presence of 10 nmol/L MBG resulted in a five-fold reduction in the levels of Fli1 and a concomitant increase in the levels of collagen-1, not affecting levels of expression of TGF-β and collagen-4. Umbilical artery rings pretreated with MBG and with vehicle exhibited similar sensitivity to a submaximal (100 nmol/L) concentration of endothelin-1 (Figure 5). In umbilical artery rings pretreated with vehicle and precontracted with 100 nmol/L endothelin-1, EC_50_ for vasorelaxant effect of sodium nitroprusside was 60 ± 15 nmol/L. Pretreatment with 10 nmol/L MBG caused a decrease in the sensitivity of precontracted umbilical artery rings to sodium nitroprusside 1.1 ± 30 μmol/L, *p* < 0.01), whereas pretreatment with 10 nmo/L MBG in the presence of canrenone markedly improved the responsiveness of vascular rings to the relaxant effect of sodium nitroprusside (EC_50_ = 7.9 ± 2.8 nmol/L, *p* < 0.01) (Figure 6).

## 3. Discussion

The main observations of the present study are that canrenone disrupts MBG signaling and restores vasorelaxation impaired by MBG in human PE. Moreover, the incubation of normal umbilical arteries with MBG, which mimics PE, blunts the relaxation of vascular rings with sodium nitroprusside, and similar to PE canrenone improves relaxation of normal vascular rings pretreated with MBG. In both scenarios, in PE and after MBG treatment, Fli1 but not TGFβ appears to be responsible for profibrotic signaling in umbilical arteries. Notably, the sample size in the vasorelaxation experiments was small, while other clinical and laboratory covariates were not taken into account, suggesting the existence of the risk of residual confounding variables.

An idea that spironolactone may function as a digitalis antagonist was first proposed by Selye who demonstrated that this drug attenuates the toxic effect of digitoxin in rat myocardium [20,21]. Subsequently, Garay et al. showed that canrenone, a metabolite of spironolactone, restored Na/K-ATPase which was inhibited in hypertensive human erythrocytes by CTS [22]. Several papers which point out that spironolactone and canrenone act as a mixed agonist/antagonist of CTS and Na/K-ATPase were summarized by Semplicini et al. [23]. The recent discovery of sodium pump signaling functions has added a new dimension to the study of CTS and fibrosis, that is, demonstrated the significance of the Na/K-ATPase “signaling mechanism” [24]. 

PE is a severe complication of pregnancy characterized by stable hypertension, proteinuria, and initial symptoms of cerebral edema. The mechanism of occurrence of PE includes the remodeling of the spiral uterine arteries and the development of fibrosis due to the release of humoral factor CTS [25,26]. Plasma levels of MBG were significantly elevated in the present study which occurred in parallel with the pressor response, with the reduction of Fli1 and elevation of Col1 in aortic membranes. Moreover, in PE MBG could come from the placenta which is one of its sites of biosynthesis [27]. In addition to MBG-induced Fli1-dependent fibrosis, we also observed the absence of the activation of another pathway, TGF-β-dependent [28]. Notably, in the present study levels of TGF-β and of collagen-4 in the aorta did not change, which indicates that this pathway is not involved in the development of vascular fibrosis in PE [28].

The growth and survival of embryos during gestation depend on the proper development and integrity of the vascular system, and fetal dysfunctions are often caused by defective angiogenesis. Fli1 deficiency is a key factor regulating a broad spectrum of endothelial behavior and vascular remodeling associated with gestation [13,28,29]. The Na/K-ATPase/Src/EGFR complex activates signaling which results in the phosphorylation of PKCδ and its translocation to the nucleus [1]. In the nucleus, phosphorylated PKCδ in turn phosphorylates Fli1, a negative regulator of collagen synthesis, and phosphorylated Fli1 withdraws from the Col1 promoter which rises expression of procollagen and collagen-1 [29].

For 20 years, Fli1 was viewed as a member of the ETS transcription factor family originally identified in erythroleukemia induced by the Friend Murine leukemia virus [30,31,32], but recent evidence indicates that Fli1 is a master of transcription factor which promotes vascular morphogenesis and angiogenesis [33,34,35]. Our present findings are in agreement with this notion. In human tissues, it has been shown that the placental levels of Fli1 were dramatically lower, while collagen-1 levels were higher in both placenta and umbilical arteries in PE samples compared to the tissues from control non-complicated pregnancy [12]. Accordingly, the subsequent treatment of the healthy human umbilical artery explants from PE patients with anti-MBG monoclonal antibody was accompanied by a significant decrease in collagen-1 abundance [12]. In agreement with our previous data, the present paper demonstrates that the MBG-Na/K-ATPase-Fli1-collagen-1 system is implicated in the pathogenesis of human PE, and indicates that the mineralocorticoid receptor antagonist, canrenone, is capable of offsetting the vasoconstrictor action of MBG. Antagonism of this deleterious effect of MBG by canrenone may be one of the new possibilities of pharmacological therapy and prevention of vascular fibrosis in conditions different from PE. For example, Fli1-dependent vascular fibrosis is a hallmark of renal failure in partially nephrectomized rats and in patients with end-stage renal disease [36]. Most recently in a group of patients with end-stage kidney disease it was found that left ventricular mass index was an independent predictor of plasma MBG [37]. In young hypertensive patients and older patients with resistant hypertension, increased pulse wave velocity is associated with increased levels of MBG in the presence of unchanged arterial pressure [7,38]. Thus, the failure of blood vessels to adequately relax may be in part mediated by MBG, even when the blood pressure is normal. Therefore, we have to consider antagonists of aldosterone as potential antagonists of cardiotonic steroids. 

## 4. Methods

### 4.1. General

The protocol for the human study was approved by the Research Council of St Petersburg School of Pediatric Medicine, Russia, and by the Institutional Review Board of Medstar Research Institute, Washington, DC, renewal on 10 March 2020 (Protocol №3/6). Twelve PE participants were enrolled in the study after giving informed consent. Diagnosis of PE was based on the criteria of the American Congress of Obstetrics and Gynecology [19]. This definition includes systolic blood pressure of at least 140 mmHg, or diastolic blood pressure of at least 90 mmHg, and new onset proteinuria (urinary protein excretion more than 0.3 g/24 h or a urinary protein concentration of more than 1 g/L in at least two random urine specimens collected 6 h or more apart in a pregnancy after the 20th week of gestation). Exclusion criteria were a clinical need for digitalis drugs, antecedent history of essential hypertension, and chronic cardiovascular, renal, hepatic or endocrine disorders. In addition, 10 age-matched and gestational age-matched normotensive subjects with uncomplicated pregnancies were enrolled to the study to serve as the control group.

### 4.2. Preparation of the Placentae Sections

The pieces of the placenta were fixed for 72 h (4 °C) in 4% p-formaldehyde solution (Sigma, St. Louis, MO, USA), washed by 0.9% Na-phosphate buffer (PBS) and immersed in PBS containing 30% sucrose at 4 °C. The pieces were frozen on dry ice using Tissue-Tek^®^ medium (Sacura Finetek Europe, AV Alphen aan den Rijn, The Netherlands) and stored at −80 °C. The sections from different levels of the pieces (10 μm) were prepared using a Leica CM-1520 cryostat (Leica Microsystems, Wetzlar, Germany) and mounted on the SuperFrost/plus glasses (Menzel, Germany), dried overnight at room temperature and used for histological or immunohistochemical analysis.

### 4.3. Histological Analysis and Immunohistochemical Identification of MBG in Placenta

For morphological study, the placenta sections were washed in PBS and stained with hematoxylin (Ehrlich) and eosin (G) according to the standard procedure and after washing in PBS a cover glass using glycerol. MBG-immunopositive material was detected in sections of the placenta. Sections were preliminarily washed in PBS, then endogenous peroxidase was blocked with 0.6% hydrogen peroxide diluted in PBS for 30 min. Then the sections were washed in PBS with 0.1% TRITON X-100 (PBST) and incubated for 1 h in a 5% blocking solution (3% goat serum and 2% bovine serum albumin dissolved in PBST). Then, the sections were incubated overnight at room temperature with primary rabbit polyclonal antibodies to MBG diluted 1:500 in 2% blocking solution. The slides with sections were washed for 30 min in three portions of PBST and incubated for 1 h with secondary goat antibodies against rabbit Ig conjugated with biotin (VectorLabs., London, UK) at a dilution of 1:600. After washing in PBS, a solution of streptavidin-peroxidase (Sigma, USA; 1:700 in PBS) was applied to the sections for 1 h, washed in PBS, and incubated in 0.05% diaminobenzidine solution with 0.03% hydrogen peroxide in PBS. After washing in distilled water, the sections were enclosed under a coverslip with glycerol. The specificity of the reaction was assessed using a negative control (reactions without primary or secondary antibodies). 

### 4.4. Microscopy

The histological and immunohistochemical reactions in placenta slides were analyzed using a Carl Zeiss Imager A1 microscope (Germany) and Carl Zeiss software (Axio Vision 4.7.2). The optical density was quantified using the Image J NIH Analysis software (NIH, Bethesda, MD, USA). In each micrograph, 2–3 areas of cytotrophoblast chorionic villus were analyzed, and optical density of MBG-immunopositive material was determined, which is expressed in arbitrary units (a.u.).

### 4.5. Umbilical Arteries

After delivery, umbilical arteries were separated from surrounding tissues and either immediately tested for contractile/relaxant properties (below) [5] or processed for determination of Fli1, collagen-1, collagen-4 and TGF-β levels. Umbilical arteries from subjects with uncomplicated pregnancies were also treated ex vivo with MBG to mimic the effects of preeclampsia. Explants of umbilical arteries from subjects with uncomplicated pregnancies were placed in Dulbecco’s Modified Eagle Medium supplemented with high glucose, glutamine, and sodium pyruvate (25 mg/kg gentamicin) (Invitrogen, Waltham, MA, USA). Arterial explants were incubated for 24 h in a 5% CO_2_ atmosphere at 37 °C in the presence of MBG (10 nmol/L) or vehicle (control).

#### 4.5.1. Isolated Umbilical Artery Contractile Studies

Endothelium-denuded rings of umbilical arteries (2.5–4.0 mm wide) were suspended at a resting tension of 3.0 g in a 15 mL organ bath (Ugo Basile, Gemonio, Italy) and superfused at 37 °C with a solution containing mmol/L: NaCl 130, KCl 4.0, CaCl_2_ 1.8, MgCl_2_ 1.0, NaH_2_PO_4_ 0.4, NaHCO_3_ 19 and glucose 5.4, and gassed with a mixture of 95% O_2_ and 5% CO_2_ (pH 7.45), and isometric contractions were recorded as reported previously [5]. The arterial rings were constricted twice with 80 mmol/L KCl, and after complete relaxation contractile responses to endothelin-1 (0.1–10 nmol/L) were studied. In parallel experiments, vasorelaxant effect of sodium nitroprusside (SNP) (1 nmol/L–1 umol/L) was studied following constriction of umbilical artery rings with 100 nmol/L endothelin-1.

#### 4.5.2. MBG Measurement

For measurement of MBG plasma and placental samples were processed as described previously in detail [11]. The assay is based on competition between immobilized antigen (MBG-glycoside-thyroglobulin) and MBG, other cross-reactants, or endogenous CTS within the sample for a limited number of binding sites on anti-MBG 4G4 monoclonal antibodies. Secondary (goat anti-mouse) antibody labeled with nonradioactive europium were obtained from PerkinElmer (Waltham, MA, USA). 

### 4.6. Western Blotting

The umbilical arteries were homogenized in RIPA buffer (Santa Cruz Biotechnology, Inc., Santa Cruz, CA, USA). Solubilized proteins were separated by 10 or 12% Tris-glycine polyacrylamide gel electrophoresis and transferred to a nitrocellulose membrane (GE Health Care/Life Sciences, Pittsburgh, PA, USA). The proteins were visualized using Fli1 (anti-rabbit polyclonal, 1:500, Santa Cruz Biotechnology Inc., Santa Cruz, CA, USA), Collagen-1 (anti-goat polyclonal, 1:500, Southern Biotechnology, Birmingham, AL, USA, 1:200), TGF-beta (TGF-β rabbit 1:500, Cell Signaling Technology, Danvers, MA, USA) and Collagen-4 (anti-mouse 1:500, Southern Biotechnology, Birmingham, UK). To normalize the levels of proteins against the levels of glyceraldehydes-3-phosphate dehydrogenase (GAPDH), the membranes were stripped and re-probed with a mouse monoclonal antibody against GAPDH (Santa Cruz Buitechnology, 1:1000) followed by anti-mouse peroxidase-conjugated antiserum (Amersham Corp., 1:2000). Immunostained proteins were visualized by 1–20 min exposure of nitrocellulose membrane with an ECL detection system (GERPN2235) (GE Health Care/Life Sciences, UK) on green-sensitive X-ray film (CEA, Göteborg, Sweden). The optical density of bands was quantified by densitometric analysis using the Image Lab program (BioRad, Hercules, CA, USA). 

### 4.7. The RNA Extraction, Reverse Transcription and Real-Time Quantitative PCR Analysis

The frozen vessels were cut into small pieces (at −15 °C) and homogenized in RNA-extract Reagent (TRIZOL-reagent, Molecular Probes Inc., Eugene, OR, USA) according to the manufacturer’s protocol for total RNA extraction. The quality and yield of the RNA were estimated by measuring the A260/A280 ratio (Nanophotometer C40, Implen, München, Germany). The samples containing 1 μg of RNA were reverse-transcribed to cDNA using the random oligodeoxynucleotide primers and the MMLV RT kit (Evrogen, Moscow, Russia) according to the manufacturer’s protocol. The amplification was performed using the mixture (final volume of 25 μL) containing 10 ng of the reverse transcription product, 0.4 μM each of the forward and reverse primers, and the qPCRmix-HS SYBR + LowROX kit (Evrogen, Russia), as described by us earlier [39]. The amplified signals were detected with the Applied Biosystems^®^ 7500 Real-Time PCR System (Life Technologies, Thermo Fisher Scientific Inc., Waltham, MA, USA). The real-time quantitative PCR amplification protocol was used: (i) initial denaturation at 95 °C for 5 min; (ii) three-stage amplification and quantification program consisting of 38 cycles of 95 °C for 30 s, 55–58 °C for 10 s and 72 °C for 30 s; and (iii) use of the ABI Melt Curve program to verify the presence of a single peak and the absence of primer-dimer formation in each template-containing reaction. The annealing temperatures were optimized using the Primer-Blast program (http://www.ncbi.nlm.nih.gov/tools/primer-blast/, accessed on 16 February 2012). In the preliminary studies, the SYBR Green-labeled PCR products were evaluated by agarose gel electrophoresis, and the authenticity of each amplicon was verified by nucleic acid sequencing. For reactions, the Forvard (F) and Revers (R) primers were used: for gene *FLI1* (*NM_002017.5*): F—CCAACGAGAGGAGAGTCATCG and R—TTCCGTGTTGTAGAGGGTGGT. The expression of the gene *GAPDH* (NM_001289745.3) was used as a housekeeping gene (F—ACAGTCAGCCGCATCTTCTT and R—GACTCCGACCTTCACCTTCC). The data were calculated using the delta–delta-C_t_ method and expressed as fold expressions relative to the expression in control patients.

### 4.8. Statistics

Data were analyzed using one-way ANOVA followed by the Bonferroni test, and by two-tailed *t*-test and Shapiro–Wilk test (when applicable) (GraphPad Prism software, San Diego, CA, USA). A *p* value of less than 0.05 was considered to be statistically significant.

## Figures and Tables

**Figure 1 ijms-23-03336-f001:**
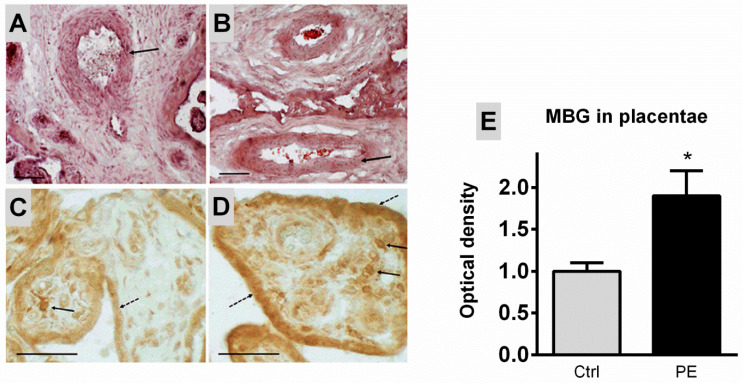
Arteries in the chorionic villi of the placenta in healthy pregnant subjects (**A**) and patients with PE (**B**). Arrows point to the muscular layer of the arteries. Stained with hematoxylin and eosin, scale bars 200 µm. Immunohistochemical staining with anti-MBG antibodies in the placenta of healthy subjects (**C**) and patients with PE (**D**); results of quantitative analysis of the optical density of MBG in the cytotrophoblast of the chorionic villi (**E**). Solid arrows—the reaction in the cells of the chorionic villi, dashed arrows—the reaction in the trophoblast of the chorionic villi. In panel E–data are presented in arbitrary units (a.u.), *—the reliability of the difference from the control (*p* < 0.05). Scale 200 µm (**A**,**B**), 100 µm (**C**,**D**).

**Figure 2 ijms-23-03336-f002:**
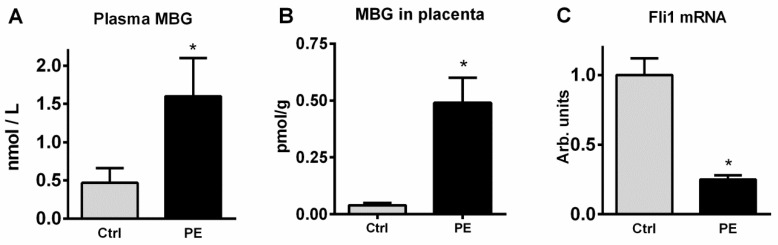
Plasma (**A**) and placental (**B**) levels of MBG, and mRNA levels of Fli1 in placenta (**C**) in subjects with uncomplicated pregnancy (Ctrl) and from patients with preeclampsia (PE). By two-way Student’s test: * *p* < 0.01 vs. Ctrl.

**Figure 3 ijms-23-03336-f003:**
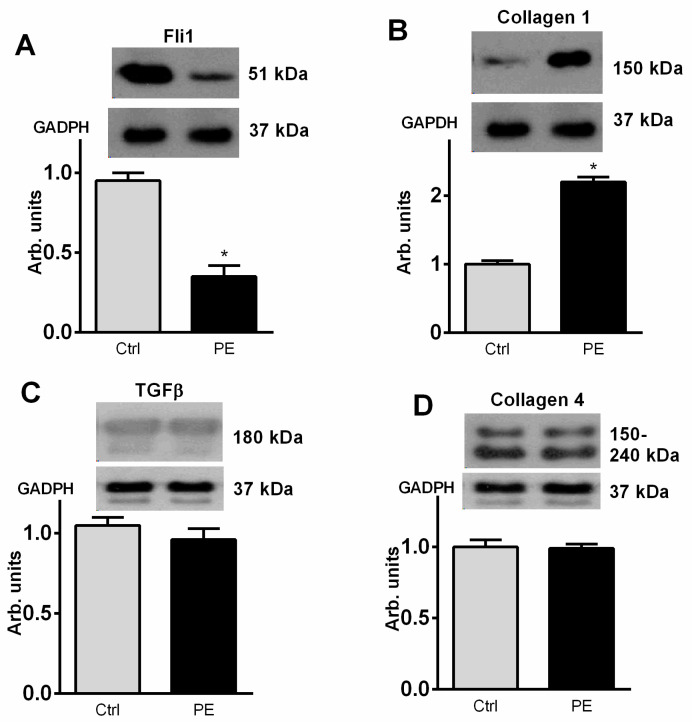
Levels of Fli1 (**A**), collagen-1 (**B**), TGFß (**C**) and collagen 4 (**D**) (Western blot) in the umbilical arteries from subjects with uncomplicated pregnancy (Ctrl) and from patients with preeclampsia (PE). Top panels. Western blotting representative bands; bottom panels, GAPDH for statistical analysis. Each bar represents individual samples pooled and averaged to give means and SEM of 5 measurements. By one-way ANOVA followed by Bonferroni test: * *p* < 0.01 vs. Ctrl.

**Figure 4 ijms-23-03336-f004:**
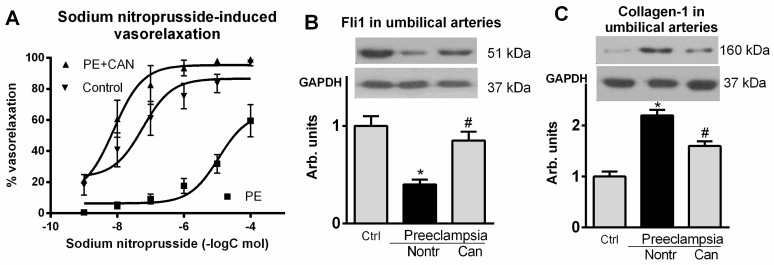
(**A**)—Effect of normotensive pregnancy, control (EC_50_ = 58 ± 14 nmol/L), PE alone (EC_50_ = 58 ± 14 nmol/L; *p* < 0.01 vs. contol) and PE with canrenone (EC_50_ = 40 ± 7 nmol/L, *p* < 0.01 in PE) on the responsiveness of umbilical artery rings to sodium nitroprusside-induced vasorelaxation following contractions induced by 100 nmol/l endothelin-1, concentration–response curves were determined 5 times and averaged to give means and SEM for IC_50_. Effects of the normotensive pregnancy (Ctrl), PE (Nontr) and PE plus canrenone (Can) on (**B**) levels of Fli1 and (**C**) collagen-1 in umbilical arteries. Top analysis. Each bar represents individual samples pooled and averaged to give means and SEM of 5 measurements. By one-way ANOVA followed by Bonferroni test: * *p* < 0.01 versus nontr. # *p* < 0.01 vs. can.

**Figure 5 ijms-23-03336-f005:**
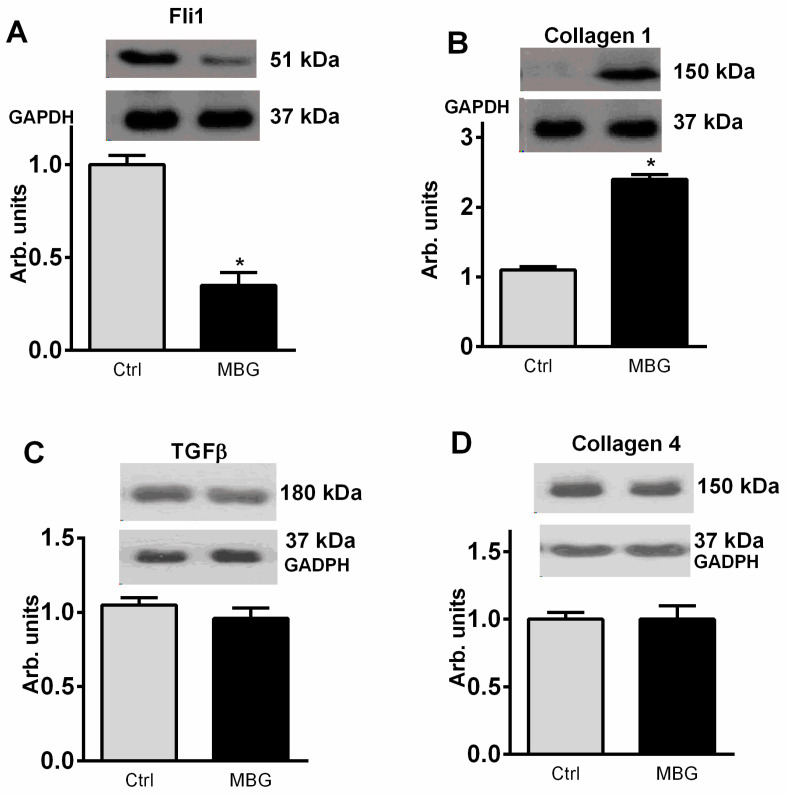
Levels of Fli1 (**A**), collagen-1 (**B**), TGFβ (**C**) and collagen 4 (**D**) by. Western blotting in the umbilical arteries from subjects with uncomplicated pregnancy nontreated (Ctrl) and treated by 10& M MBG. Top panels, Western blotting representative bands; bottom panels, GAPDH for statistical analysis. Each bar represents individual samples pooled and averaged to give means and SEM of 5 measurements. By one-way ANOVA followed by Bonferroni test: * *p* < 0.01 vs. Ctrl.

**Figure 6 ijms-23-03336-f006:**
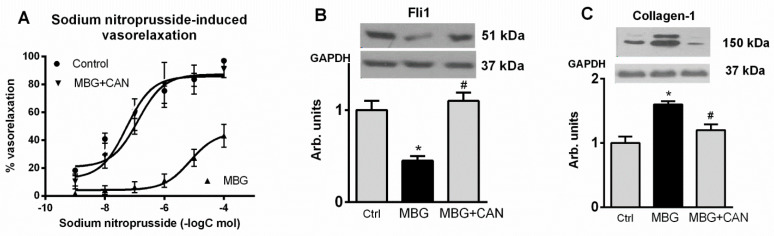
(**A**)—Effect of the normotensive pregnancy, control (60 ± 15 nmol/L), normotensive pregnancy in the presence MBG (1.1 ± 30 μmol/L; *p* < 0.01 vs. contol), and normotensive pregnancy in the presence of MBG and canrenone (EC_50_ = 7.9 ± 2.8 nmol/L, *p* < 0.01 vs. effect of MBG) on the responsiveness of umbilical artery rings to sodium nitroprusside-induced vasorelaxation following contractions induced by 100 nmol/L endothelin-1. Concentration–response curves were determined 5 times and averaged to give means and SEM for IC_50_. Effects of the normotensive pregnancy (Ctrl), PE (Nontr) and PE plus canrenone (Can) on (**B**) levels of Fli1 and (**C**) collagen-1 in umbilical arteries. Top panels, Western blotting representative bands; bottom panels, GAPDH for statistical analysis. Each bar represents individual samples pooled and averaged to give means and SEM of 5 measurements. By one-way ANOVA followed by Bonferroni test: * *p* < 0.01 versus nontr. # *p* < 0.01 vs. can.

**Table 1 ijms-23-03336-t001:** Characteristics of the study participants.

	Control Group (*n* = 12)	Patients with PE (*n* = 15)
Age (years)	34 ± 1	34 ± 2
Gestational age (weeks)	39 ± 0.5	37.9 ± 0.2
Systolic BP (mmHg)	116 ± 2	148 ± 4 *
Diastolic BP (mmHg)	73 ± 1	94 ± 3 *
Protein excretion (g per 24 h)	n/d	1.4 ± 0.3 *

Means ± SEM. By two-tailed *t*-test and by Shapiro–Wilk test: * *p* < 0.01 vs. control group. Control group, subjects with normal pregnancy; PE, patients with preeclampsia; BP, blood pressure.

## Data Availability

The data presented in this study are available on request from the corresponding author.

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
