# Peer review of "Canrenone Restores Vasorelaxation Impaired by Marinobufagenin in Human Preeclampsia"

_ijms, 2022, doi:10.3390/ijms23063336_

Round 1
Reviewer 1 Report
This study by N.I. Agalakova et al. is straightforward: Cardiotonic steroid marinobufagening (MBH) is elevated in pregnant women with preeclampia (PE) in parallel with fibrosis of umbilical arterios and impairment of relaxation by sodium nitropruside of such arteries, linked to Na+/K-ATPase. Furthermore canrenone, the active metabolite of spironolactone, a mineralcorticoid antagonist, decrease the fibrotic effect of MBG and restored the vascular relaxation. Of interest was the observation that this counterbalance effect was observation that this counterbalance effect was observed in both umbilical arteries from PE women and in normal arteries incubated for 24 h with MBG. Authors suggest that cancerone is capable of reverting the vascoconstrictor action MBG and hence, the drug exhibits therapeutic potential in pregunt women with pE.
This study emanates for similar approuches on other vessels and cardiovascular conditions, from the laboratory of those authors. The present manuscript is clearly written, and data are rigorously present with appropriate statistical analysis.
Author Response
We are most grateful to the Reviewer for his time.
Reviewer 2 Report
The present study evaluated the role of marinobufagenin in preeclampsia. The following comments should be taken into account.
- The statistical methodology should be described in more detail. Specifically, how was the normality tested?
- A more complete hypothesis about the pathogenetic implications of the study should be presented in the Discussion section.
- The limitations of the study should be described in more detail. For example, it is important to note that the sample size was small, while other clinical and laboratory covariates were not taken into account. Therefore, the risk of residual confounding should not be ignored.
- The clinical implications of the study's findings should be explained in more detail.
Author Response
We are most grateful to the Reviewers for their time and insightful critique. We agree with the Reviewers’ comments and we have adjusted the manuscript along the lines they have suggested.
- The statistical methodology should be described in more detail. Specifically, how was the normality tested?
The normality was evaluated by Shapiro-Wilk test (GraphPad Prism software; GraphPad Inc.) was performed. All samples passed the normality test, and systolic and diastolic blood pressure and gestational age were not normally distributed. Thank you for bringing that up.
- A more complete hypothesis about the pathogenetic implications of the study should be presented in the Discussion section.
This has been done. Discussion, last paragraph.
- The limitations of the study should be described in more detail. For example, it is important to note that the sample size was small, while other clinical and laboratory covariates were not taken into account. Therefore, the risk of residual confounding should not be ignored.
A senrence describing limitations is added. Discussion, end of first paragraph.
- The clinical implications of the study's findings should be explained in more detail.
This has been done. Discussion, last paragraph.